# VidEdit: Zero-Shot and Spatially Aware Text-Driven Video Editing

**Paul Couairon**                                                        *paul.couairon@isir.upmc.fr*
*Thales SIX GTS France, ThereSIS Lab, Palaiseau, France*
*Sorbonne Université, CNRS, ISIR, F-75005 Paris, France*

**Clément Rambour**                                                      *clement.rambour@cnam.fr*
*Cnam, CEDRIC, Paris, 75003, France.*

**Jean-Emmanuel Haugeard**                                   *jean-emmanuel.haugeard@thalesgroup.com*
*Thales SIX GTS France, ThereSIS Lab, Palaiseau, France*

**Nicolas Thome**                                                        *nicolas.thome@isir.upmc.fr*
*Sorbonne Université, CNRS, ISIR, F-75005 Paris, France*

**Reviewed on OpenReview:** *https://openreview.net/forum?id=i02A009I5a*

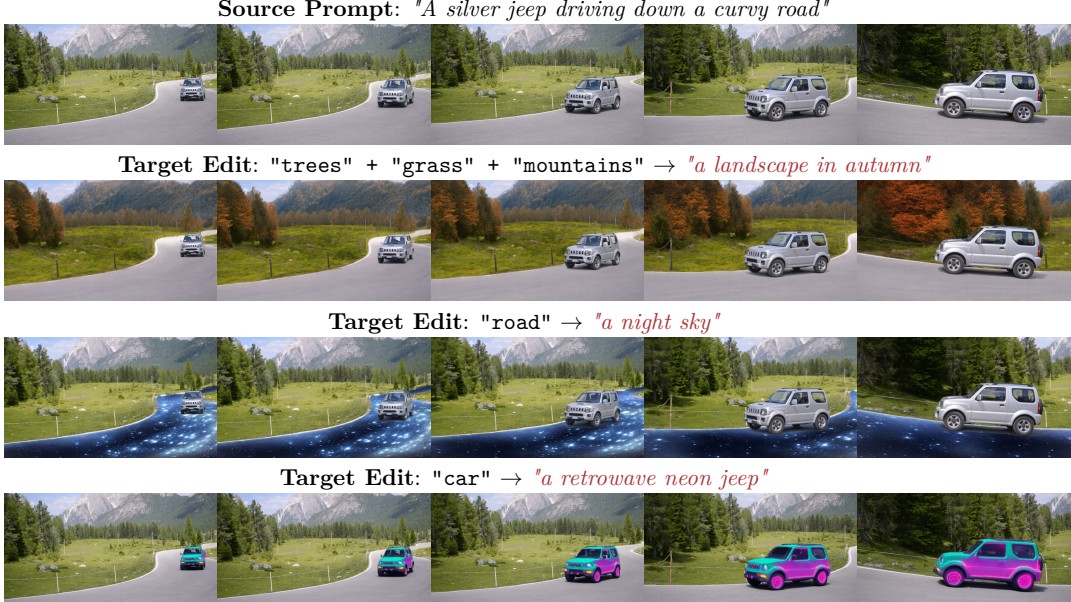

Figure 1: **VidEdit** allows to perform rich and diverse video edits on a precise semantic region of interest while perfectly preserving untargeted areas. The method is lightweight and maintains a strong temporal consistency on long-term videos.

## Abstract

Recently, diffusion-based generative models have achieved remarkable success for image generation and edition. However, existing diffusion-based video editing approaches lack the ability to offer precise control over generated content that maintains temporal consistency in long-term videos. On the other hand, atlas-based methods provide strong temporal consistency but are costly to edit a video and lack spatial control. In this work, we

introduce VIDEDIT, a novel method for zero-shot text-based video editing that guarantees robust temporal and spatial consistency. In particular, we combine an atlas-based video representation with a pre-trained text-to-image diffusion model to provide a training-free and efficient video editing method, which by design fulfills temporal smoothness. To grant precise user control over generated content, we utilize conditional information extracted from off-the-shelf panoptic segmenters and edge detectors which guides the diffusion sampling process. This method ensures a fine spatial control on targeted regions while strictly preserving the structure of the original video. Our quantitative and qualitative experiments show that VIDEDIT outperforms state-of-the-art methods on DAVIS dataset, regarding semantic faithfulness, image preservation, and temporal consistency metrics. With this framework, processing a single video only takes approximately one minute, and it can generate multiple compatible edits based on a unique text prompt.

# 1 Introduction

Diffusion-based models (Ho et al., 2020; Song et al., 2020; Rombach et al., 2022; Ramesh et al., 2022) have recently taken over image generation. In contrast to generative adversarial networks (Goodfellow et al., 2020; Karras et al., 2018; Yu et al., 2021) which are notoriously difficult to train, diffusion models offer a more reliable training process and consistently generate highly convincing samples. Besides, they can also be used for editing purposes by integrating conditional modalities such as text (Rombach et al., 2022), edge maps or beyond (Zhang & Agrawala, 2023; Mou et al., 2023). Such capacities have given rise to numerous methods that assist artists in their content creation endeavor (Tumanyan et al., 2022; Kawar et al., 2022).

Yet, unlike image editing, text-based video editing represents a whole new challenge. Indeed, naive frame-wise application of text-driven diffusion models leads to flickering video results that look poor to the human eye as they lack motion information and 3D shape understanding. To overcome this challenge, numerous methods introduce diverse spatiotemporal attention mechanisms that aim to preserve objects' appearance across neighboring frames while respecting the motion dynamics (Wu et al., 2022; Qi et al., 2023; Ceylan et al., 2023; Wang et al., 2023; Liu et al., 2023). However, they not only demand significant memory resources but also concentrate on a limited number of frames as the proposed spatiotemporal attention mechanisms are not reliable enough over time to model long-term dependencies. On the other hand, current atlas-based video editing methods (Bar-Tal et al., 2022; Loeschcke et al., 2022) require costly optimization procedures for each text query and do not enable precise spatial editing control nor produce diverse samples.

This paper introduces VIDEDIT, a simple and effective zero-shot text-based video editing method that shows high temporal consistency and offers object-level control over the appearance of the video content. The rationale of the approach is shown in Fig 1. Given an input video and a target edit, *e.g.* `"road"` → *"a night sky"*, VidEdit precisely delineates the region of interest in the atlas space as well as the internal edges that characterize its semantic structure. The text prompt and the edge map are then passed to a pre-trained conditional diffusion model that generates an edit that matches these controls. During the generation phase, the edit seamlessly merges with the original atlas through a blended diffusion process (Avrahami et al., 2022), which leaves the remainder of the video content unchanged. We hypothesize and confirm that diffusion models can effectively handle distortions in atlases, allowing us to modify these representations with little effort. Furthermore, by directing the generation process with conditional inputs, we can create compelling video edits with a fine-grained spatial control that maintain temporal consistency. To achieve this goal, the approach includes two main contributions.

Firstly, we combine the strengths of atlas-based approaches and text-to-image diffusion models. The idea is to decompose videos into a set of layered neural atlases (Kasten et al., 2021) which are designed to provide an interpretable and semantic unified representation of the content. We then apply a pre-trained text-driven image diffusion model to perform zero-shot atlas editing, the temporal coherence being preserved when edits are mapped back to the original frames. Consequently, the approach is *training free* and efficient as it can edit a full video in about one minute. In addition, we take special care to preserve the structure and geometry in the atlas space as it not only encodes objects' temporal appearance but also their movements and spatial placement in the image space. Therefore, to constrain the edits to match as accurately as possible the semantic

layout of an atlas representation, we leverage an off-the-shelf panoptic segmenter (Cheng et al., 2022) as well as an edge detection model (HED) (Xie & Tu, 2015). The segmenter extracts the regions of interest whereas the HED specifies the inner and outer edges that guide the editing process for an optimal video content alteration/preservation trade-off. Hence, we adapt the utilization of a spatially grounded editing method to a conditional diffusion process that operates on atlas representations. This is achieved by extracting a crop around the area of interest and intentionally utilizing a non-invertible noising process.

We conduct extensive experiments on DAVIS dataset, providing quantitative and qualitative comparisons with respect to video baselines based or not on atlas representations, and frame-based editing methods. We show that VIDEDIT outperforms these baselines in terms of semantic matching to the target text query, original content preservation, and temporal consistency. Especially, we highlight the benefits of our approach for foreground or background object editions. We also illustrate the importance of the proposed contributions for optimal performance. Finally, we show the efficiency of VIDEDIT and its capacity to generate diverse samples compatible with a given text prompt.

## 2 Related Work

**Text-driven Image Editing.** In the past few years, Text-to-Image (T2I) generation has become an increasingly hot topic. Recently, these generative models have benefited from the swelling popularity of diffusion models (Ho et al., 2020; Song et al., 2020; Ramesh et al., 2022) as well as the accurate image-text alignment provided by CLIP (Radford et al., 2021). Latent Diffusion Models (LDMs) (Rombach et al., 2022) propose to enhance the training efficiency, memory, and runtime of such models, by taking the diffusion process into the latent space of an autoencoder. As a result, they have taken over text-driven image generation and editing. For example, SDEdit (Meng et al., 2021) proposes to corrupt an image by adding Gaussian noise, and a text-conditioned diffusion network denoises it to generate new content. Other works aim to perform local image editing by using an edit mask (Avrahami et al., 2022; Couairon et al., 2022) and combining the features of each step in the generation process for image blending. Still focusing on image-to-image translation, (Hertz et al., 2022) or (Tumanyan et al., 2022) extract attention features to constrain the editions to regions of interest. Kawar et al. (2022) or Mokady et al. (2022) refine image editing via an optimization procedure.

**Text-driven Video Editing.** While significant advances have been made in T2I generation, modeling strong temporal consistency for video generation and editing is still a labor in progress. Numerous works aim to generate original video content directly from an input text query with novel spatiotemporal attention mechanisms (Ho et al., 2022b;a; Villegas et al., 2022; Singer et al., 2022; Zhou et al., 2022). However, these methods still exhibit flickering artifacts and inconsistencies that alter the quality of the visual outputs. When it comes to video editing, existing approaches can be categorized into two main groups. On one side are methods that seek to adapt the structure of a frozen T2I diffusion model to perform video editing in a zero-shot manner. Tune-A-Video (Wu et al., 2022) overfits a video on a given text query and generate new content from similar prompts. Other approaches (Liu et al., 2023; Qi et al., 2023; Shin et al., 2023; Ceylan et al., 2023; Wang et al., 2023) propose spatiotemporal attention mechanisms to transfer pre-trained T2I model knowledge to text-to-video. However, these methods still struggle to ensure reliable long-term coherence. On the other side, Neural Layered Atlases (Kasten et al. (2021)) provides a method for decomposing video content into a set of 2D atlases that can be edited and mapped back to the frame space, ensuring excellent temporal consistency. Based on such atlases, Text2Live (Bar-Tal et al. (2022)) facilitates coherent text-to-video editing by optimizing an edit layer over the atlas. However, its costly optimization for each prompt limits its ability to generate edits on the fly. VIDEDIT follows the trail set by Text2Live in atlas-based video editing by harnessing the adaptability and efficiency of a pre-trained T2I diffusion model to perform atlas editing. We thereby eliminate any optimization procedure and enable precise user-control and quick inference.

## 3 VidEdit Framework

The high visual quality offered by T2I diffusion models as well as their effectiveness to generate samples that are aligned with provided conditional information motivate us to utilize these models to perform our video

editing task in the 2D atlas space. To this end, we introduce VIDEDIT, a novel lightweight, and consistent video editing framework that provides object-scale control over the video content. The main steps of VIDEDIT are illustrated in Fig. 2. First, we propose to benefit from Neural Layered Atlas (NLA) (Kasten et al., 2021) to build global representations of the video content ensuring strong spatial and temporal coherence. Second, the underlying global scene encoded in the atlas representation is processed through a zero-shot image editing diffusion procedure. Text-based editing inherently faces the difficulty of accurately identifying the region to edit from the input text and may as well deteriorate neighboring regions or introduce rough deformations in the object aspect (Wu et al., 2022; Ceylan et al., 2023; Liu et al., 2023). We avoid these pitfalls by carefully extracting rich semantic information using HED maps and off-the-shelf segmentation models (Cheng et al., 2022) to guide the diffusion generative process. We adapt their design and utilization for atlas images.

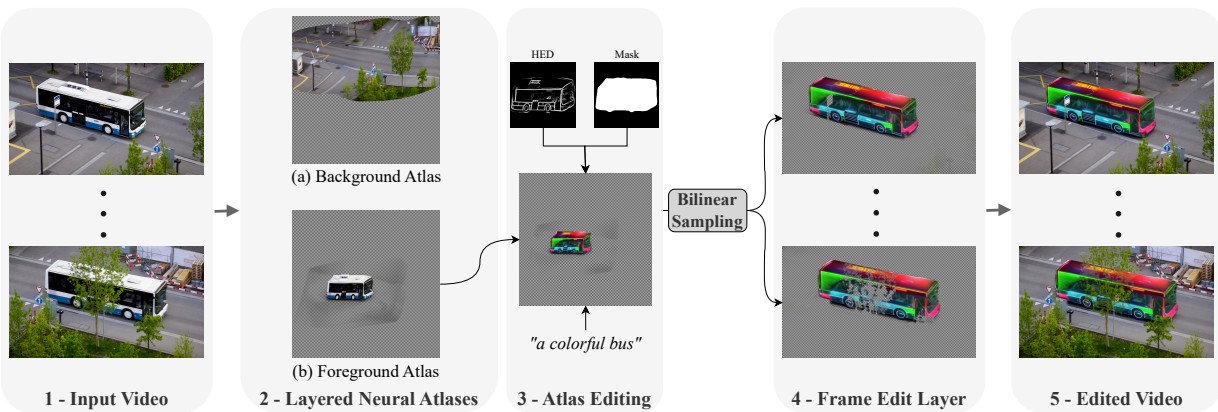

Figure 2: **Our VidEdit pipeline**: An input video **(1)** is fed into NLA models (Kasten et al., 2021) which learn to decompose it into 2D atlases **(2)**. Depending on the object we want to edit, we select an atlas representation onto which we apply our editing diffusion pipeline **(3)**. The edited atlas is then mapped back to frames via a bilinear sampling from the associated pre-trained network $\mathbb{M}$ **(4)**. Finally, the frame edit layers are composited over the original frames to obtain our desired edited video **(5)**.

## 3.1 Zero-shot Atlas-based video editing

**Neural Layered Atlases**. Neural Layered Atlases (NLA) (Kasten et al., 2021) provide a unified 2D representation of the appearance of an object or the background through time, by decomposing a video into a set of 2D atlases. Formally, each pixel location $p = (x, y, t) \in \mathbb{R}^3$ is fed into three mapping networks. While $\mathbb{M}_f$ and $\mathbb{M}_b$ map $p$ to a 2D $(u, v)$-coordinate in the foreground and background atlas regions respectively, $\mathbb{M}_\alpha$ predicts a foreground opacity value:

$$\mathbb{M}_b(p) = (u_b^p, v_b^p), \qquad \mathbb{M}_f(p) = (u_f^p, v_f^p), \qquad \mathbb{M}_\alpha(p) = \alpha^p \tag{1}$$

Each of the predicted $(u, v)$-coordinates are then fed into an atlas network $\mathbb{A}$, which yields an RGB color at that location. Color can then be reconstructed by alpha-blending the predicted foreground $c_f^p$ and background $c_b^p$ colors at each position $p$, according to the corresponding opacity value $\alpha^p$:

$$c^p = (1 - \alpha^p)c_b^p + \alpha^p c_f^p. \tag{2}$$

We train NLA in a self-supervised manner as in Kasten et al. (2021). The obtained background and foreground atlases are large 2D pixel representations disentangling the layers from the video. By utilizing these mapping and opacity networks, one can edit the RGBA pixel values and project them back onto the original video frames.

**Zero-shot atlas editing.** The 2D atlases obtained by disentangling the video are a well-posed framework to edit objects while ensuring a strong temporal consistency. We propose here to perform zero-shot text-based editing of atlas images. This is in sharp contrast with Bar-Tal et al. (2022), which requires training a specific

generative model for each target text query. We use a pre-trained conditioned latent diffusion model, although our approach is agnostic to the image editing tool. As illustrated with Fig. 2, the automatic video editing task is transformed into a much straightforward, training free, and adaptable image editing task, resulting in competitive performance.

## 3.2 Semantic Atlas Editing with VidEdit

2D atlas representations pave the way to use powerful off-the-shelf segmentation models (Xu et al., 2023; Kirillov et al., 2023; Zou et al., 2023) to precisely circumscribe the regions-of-interest. The results are then clean object-level editions maximizing the consistency with the original video and the rendering of the targeted object. In addition, we also extract HED maps as they lead to rich object descriptions. We then use the extracted masks to guide the generative process of a DDIM (Denoising Diffusion Implicit Model) model conditioned by both a target prompt and a HED map, the latter ensuring to preserve the semantic structure of the source image. The whole pipeline is illustrated in Fig. 3.

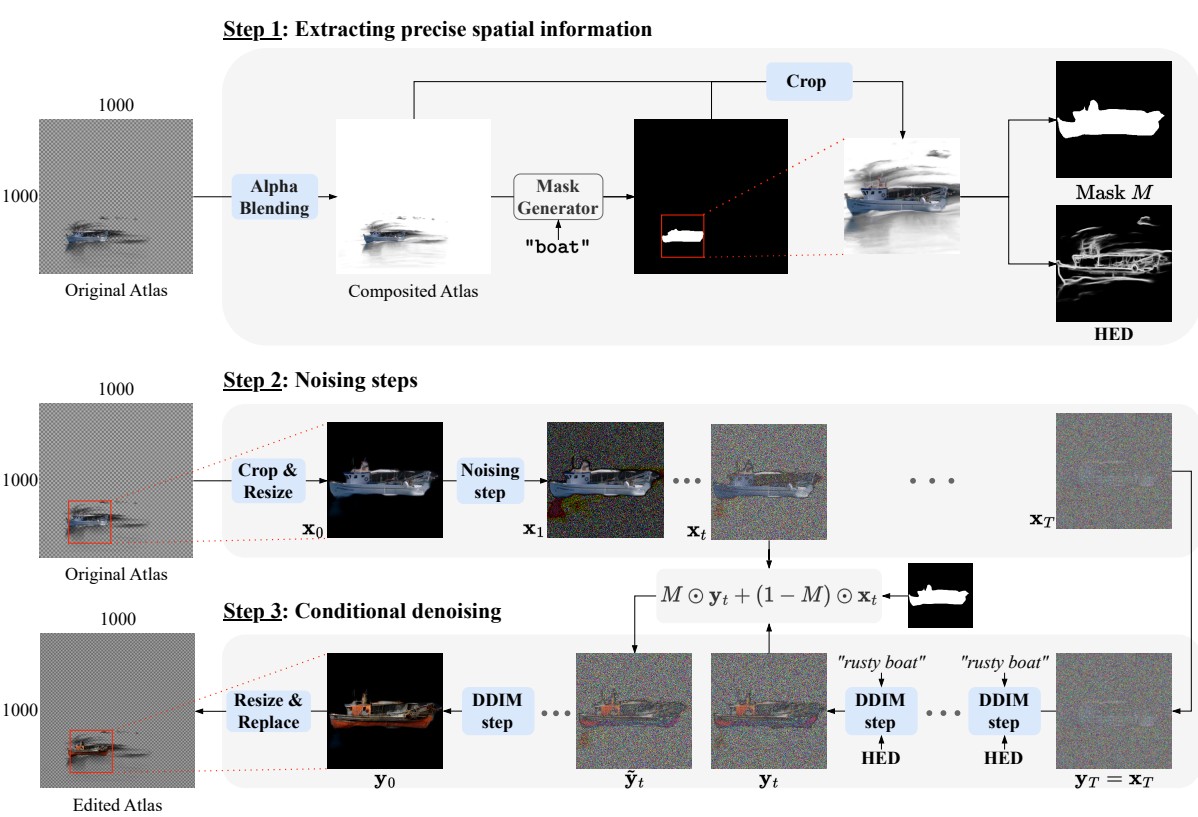

Figure 3: **The three steps of our atlas editing procedure.**

**Step 1: Extracting precise spatial information.** In order to generate edits that are meaningful and realistic once mapped back in the original image space, we have to guide the generative process toward a plausible output in the atlas representation. Our objective is then twofold. First, we want to precisely localize our region of interest in the atlas in order to only make alterations within this area. As in Avrahami et al. (2022), this edit mask will help to seamlessly blend our edits in the video content while having minimal impact on out-of-interest parts of the video. Recently, Couairon et al. (2022) proposed a method to automatically infer such a mask with a reference and target text queries, but it generally overshoots the region that requires to be edited, compromising the integrity of the original video content. On the other hand, segmentation models have recently seen spectacular advances (Cheng et al., 2022; Xu et al., 2023; Kirillov et al., 2023; Zou et al., 2023), allowing to confidently and accurately detect and recognize objects in images. When applied directly onto atlas grids, we observe that, despite the distribution shift with real-world images, these models

generalize sufficiently well to infer a mask around the targeted regions. Consequently, we choose to leverage the performance of these frameworks to perform panoptic segmentation and thus gain object-level spatial control over our future edits. Hence, we first take our original atlas representation which is composed of an RGB image and an alpha channel. In order to assist the segmentation network in providing a precise mask, we mix the RGB image with a fully white patch according to the alpha values. This step allows to enhance the contrast between the object and the background as illustrated in Appendix C. Then, we identify the object or region that we need to locate and create a bounding box around the identified area. Finally, we produce a more accurate mask $M$ on this smaller patch.

Second, as we are interested in changing the aspect of objects while preserving their overall shapes, we have to ensure that our edits match their semantic structure in the atlas representation. Several works propose methods to perform image-to-image translation (Mokady et al., 2022; Tumanyan et al., 2022; Bar-Tal et al., 2022; Hertz et al., 2022). However, their various drawbacks in terms of editing time or lack of generalization on atlas representations that are too far away from real-world images, hinder the use of such approaches directly in the atlas space. Consequently, we choose to align the internal knowledge of a generative text-to-image model with an external control signal that helps preserving the semantic structure of objects. To this end, we opt to exploit the accurate and computationally efficient HED algorithm (Xie & Tu, 2015) to bring out critical edges that characterize the structure of our image.

**Step 2: Noising steps.** We crop a patch from the original atlas at the bounding box location. This cropped patch is then encoded into an image latent via the VQ-autoencoder of the diffusion model. Starting from this latent dubbed $\mathbf{x}_0$, we use a classical noising procedure with $T = 1000$ steps, which leads to a nearly isotropic Gaussian noise sample $\mathbf{x}_T$, *i.e.* $p_\theta(\mathbf{x}_T) = \mathcal{N}(\mathbf{0}, \mathbf{I})$. We denote $\rho$ the noising ratio of a noisy latent $\mathbf{x}_t$ such that $\rho = t/T$.

**Step 3: Decoding with mask guidance.** Starting from our latent $\mathbf{y}_T = \mathbf{x}_T$, we decode it with a pre-trained diffusion model that integrates control modalities (Zhang & Agrawala, 2023) to guide the denoising process. Specifically, at each step $t$, the U-Net denoises the image latent in a direction determined by both the target prompt and the HED edge map:

$$\mathbf{y}_{t-1} = \sqrt{\alpha_{t-1}} \left( \frac{\mathbf{y}_t - \sqrt{1 - \alpha_t} \epsilon_\theta(\mathbf{y}_t, t, \mathbf{c}_p, \mathbf{c}_h)}{\sqrt{\alpha_t}} \right) + \sqrt{1 - \alpha_{t-1}} \epsilon_\theta(\mathbf{y}_t, t, \mathbf{c}_p, \mathbf{c}_h) \tag{3}$$

where $\mathbf{c}_p$ and $\mathbf{c}_h$ are the embeddings of the query text prompt and HED map, projected into a common representation space with $\mathbf{y}_t$, through dedicated cross-attention blocks. The encoder of the denoising U-Net $\epsilon_\theta$ is applied separately on $\mathbf{y}_t$, the input to be denoised, and the HED conditioning $\mathbf{c}_h(\lambda)$ with $\lambda$ a balancing coefficient that the decoder takes at each stage to compute a weighted sum of the activation maps. $\{\alpha_t \in (0, 1)\}_{t=1}^{T}$ is a variance schedule that determines the step sizes.

The marginal of the forward process sample at step $t - 1$ admits a simple closed form given by $\mathbf{x}_{t-1} = \sqrt{\bar{\alpha}_{t-1}} \mathbf{x}_0 + \sqrt{1 - \bar{\alpha}_{t-1}} \epsilon_{t-1}$. Following Avrahami et al. (2022), we use this relation to retrieve the area outside the object's mask $M$ during the generation process while the interior region is obtained following the standard diffusion process given in Eq. (3):

$$\tilde{\mathbf{y}}_{t-1} = M \odot \mathbf{y}_{t-1} + (1 - M) \odot \mathbf{x}_{t-1} \tag{4}$$

In the last step, the entire region outside the mask is replaced with the corresponding region from the input image, allowing to preserve exactly the background from the original crop. Our edited patch is finally replaced at its location within the atlas grid. Therefore, this pipeline seamlessly fuse the edited region with the unchanged parts of an atlas. Lastly, the edited atlas is used to perform bilinear sampling of frame edit layers. Once these layers are composited with their corresponding original frames, they produce an edited video that exhibits both spatial and temporal consistency.

## 4   Experiments

In this section, we describe our experimental setup, followed by qualitative and quantitative results.

### 4.1 Experimental setup

**Dataset.** Following Bar-Tal et al. (2022); Wu et al. (2022); Qi et al. (2023), we evaluate our approach on videos from DAVIS dataset (Pont-Tuset et al., 2017) resized at a $768 \times 432$ resolution. The length of these videos ranges from 20 to 70 frames. To automatically create edit prompts, we use a captioning model (Li et al., 2022) to obtain descriptions of the original video content and we manually design 4 editing prompts for each video.

**VidEdit setup.** To control the semantic layout of our generated edits, we utilize a ControlNet variant of Stable Diffusion (Zhang & Agrawala, 2023). This model has learned to detect and to integrate HED edges as conditional information to a diffusion model via training a copy of its layers while also maintaining locked the pre-trained parameters separately. The trainable and locked copies of the parameters are connected at each block of the UNet decoder via "zero convolution" layers that are also optimized. We refer to the original paper for additional information. The original version of Stable Diffusion is trained at a $512 \times 512$ resolution on LAION-5B dataset (Schuhmann et al., 2022). We choose Mask2former (Cheng et al., 2022) as our instance segmentation network. To edit an atlas, we sample pure Gaussian noise (*i.e.* $\rho = 1$) and denoise it for 50 steps with DDIM sampling and classifier-free guidance (Ho & Salimans, 2022). For a single 70 frames video, it takes $\sim 15$ seconds to edit a $512 \times 512$ patch in an atlas and $\sim 1$ minute to reconstruct the video with the edit layer on a NVIDIA TITAN RTX, a graphic card accessible to the general public. We set up the HED strength $\lambda$ to 1 by default.

**Baselines.** We compare our method with two text-to-image frame-wise editing approaches and three text-to-video editing baselines. (1) *SDEdit* (Meng et al., 2021) is a framewise zero-shot editing approach that corrupts an input frame with noise and denoise it with a target text prompt. (2) *ControlNet* (Zhang & Agrawala, 2023) performs frame-wise editing with an external condition extracted from the target frame. (3) *Text2Live* (Bar-Tal et al., 2022) is a Neural Layered Atlas (NLA) based method that trains a generator for each text query to optimize a CLIP-based loss. (4) *Tune-a-Video* (TAV) (Wu et al., 2022) fine-tunes an inflated version of a pre-trained diffusion model on a video to produce similar content. (5) *Pix2Video* (Ceylan et al., 2023) uses a structure-guided image diffusion model to perform text-guided edits on a key frame and propagate the changes to the future frames via self-attention feature injection.

**Metrics.** A video edit is expected to (1) be temporally consistent **(Temporal)**, (2) faithfully render a target text query **(Semantics)**, (3) preserve out-of-interest regions unaltered **(Similarity)**. To evaluate CLIP based metrics, we used CLIP ViT-L/14 (Radford et al., 2021).

- **Temporal**

  - **Frame Consistency ($\mathcal{C}_{\mathbf{Frame}}$).** Measures the CLIP similarity between the image embeddings of consecutive video frames. Formally, it writes:

$$\mathcal{C}_{\text{Frame}} = \frac{1}{N} \sum_{k=1}^{N} \frac{1}{F_k} \sum_{j=1}^{F_k-1} CLIPScore(\bar{I}_j^k, \bar{I}_{j+1}^k) \tag{5}$$

    where $\bar{I}_j^k$ is the $j$-th frame of edited video $k$, $F_k$ the number of frames in video $k$ and $N$ the number of edited videos.

  - **Warping Error ($\mathcal{E}_{\mathbf{Warp}}$).** Measures the temporal stability of edited videos based on the flow warping error between two frames:

$$\mathcal{E}_{\text{Warp}} = \frac{1}{N} \sum_{k=1}^{N} \frac{1}{F_k} \sum_{j=1}^{F_k-1} Warp(\bar{I}_j^k, \bar{I}_{j+1}^k) \tag{6}$$

    with $Warp(\bar{I}_j^k, \bar{I}_{j+1}^k)$, the warping error between consecutive frames of an edited video defined as in Lai et al. (2018)

- **Semantics**

- **Prompt Consistency ($\mathcal{C}_{\mathbf{Prompt}}$).** Measures the average CLIP similarity between a target text query and each video frame. For a unique pair *(image; caption)* the CLIP similarity writes: $CLIPScore(I, C) = \max(100 \times cos(E_I, E_C), 0)$ with $E_I$ the visual CLIP embedding for an image $I$, and $E_C$ the textual CLIP embedding for a caption $C$. $\mathcal{C}_{\mathrm{Prompt}}$ then writes :

$$\mathcal{C}_{\mathrm{Prompt}} = \frac{1}{N} \sum_{k=1}^{N} \frac{1}{F_k} \sum_{j=1}^{F_k} CLIPScore(\bar{I}_j^k, \bar{C}^k) \tag{7}$$

  where $\bar{I}_j^k$ is the $j$-th frame of edited video $k$, $\bar{C}^k$ the edited caption of video $k$, $F_k$ the number of frames in video $k$ and $N$ the number of edited videos.

- **Frame Accuracy ($\mathcal{A}_{\mathbf{Frame}}$).** Corresponds to the average percentage of edited frames that have a higher CLIP similarity with the target text query than with their source caption. Formally, $\mathcal{A}_{\mathrm{Frame}}$ writes:

$$\mathcal{A}_{\mathrm{Frame}} = \frac{1}{N} \sum_{k=1}^{N} \frac{1}{F_k} \sum_{j=1}^{F_k} \mathbb{1}\{CLIPScore(\bar{I}_j^k, \bar{C}^k) > CLIPScore(\bar{I}_j^k, C^k)\} \times 100 \tag{8}$$

- **Directional Similarity ($\mathcal{S}_{\mathbf{Dir}}$).** Quantifies how closely the alterations made to an original image align with the changes between a source caption and a target caption. For the $j$-th frame of video $k$ the similarity score writes: $SIMScore(\bar{I}_j^k, I_j^k, \bar{C}^k, C^k) = 100 * cos(E_{\bar{I}_j^k} - E_{I_j^k}, E_{\bar{C}_j^k} - E_{C_j^k})$. $\mathcal{S}_{\mathrm{Dir}}$ then writes:

$$\mathcal{S}_{\mathrm{Dir}} = \frac{1}{N} \sum_{k=1}^{N} \frac{1}{F_k} \sum_{j=1}^{F_k} SIMScore\left(\bar{I}_j^k, I_j^k, \bar{C}^k, C^k\right) \tag{9}$$

- **Similarity**

  Regarding content preservation, we have chosen three metrics that operate on different feature spaces in order to capture a rich description of perceptual similarities.

  - **LPIPS** (Zhang et al., 2018) operates on the deep feature space of a VGG network and has been shown to match human perception well.

  - **HaarPSI** (Reisenhofer et al., 2018) While LPIPS evaluates the perceptual similarity between two images in a deep feature space, HaarPSI performs a Haar wavelet decomposition to assess local similarities.

  - **PSNR** measures the distance with an original image in the pixel space.

  These metrics are extensively described in the literature and we refer to it for further details.

- **Aggregate Score.** This metric synthesizes in a single score the overall performance of each model on semantic and similarity aspects, relatively to the best baseline. When dealing with metrics where a higher value is considered preferable, a coefficient in the aggregate score is computed as: $\max(\mathcal{S}_i)/\mathcal{S}_i^j$ *i.e.* the best score for metric $i$ divided by the score of baseline $j$ for metric $i$. When the objective is to minimize the metric, we take the inverse value. The minimal and best aggregate score for each aspect is 3, as we aggregate three semantic or similarity scores.

## 4.2 State-of-the-art comparison

**Quantitative results.** Tab. 1 gathers the overall comparison with respect to the chosen baselines[1]. VIDEDIT outperforms other approaches in terms of both semantic and similarity metrics. Moreover, it exhibits a temporal consistency comparable to Text2Live while largely surpassing alternative approaches. Regarding

---

[1]Results in bold correspond to the best methods based on a paired $t$-test (risk 5%).

semantic metrics, as indicated by our best directional similarity score, VIDEDIT performs highly consistent edits with respect to the change between the target text query and the source caption.

Table 1: **State-of-the-art comparison.**

| Method | Semantic | | | | Similarity | | | | Temporal | | |
|---|---|---|---|---|---|---|---|---|---|---|---|
| | $\mathcal{C}_{\text{Prompt}}(\uparrow)$ | $\mathcal{A}_{\text{Frame}}(\uparrow)$ | $\mathcal{S}_{\text{Dir}}(\uparrow)$ | Agg. Score ($\downarrow$) | LPIPS ($\downarrow$) | HaarPSI ($\uparrow$) | PSNR ($\uparrow$) | Agg. Score ($\downarrow$) | $\mathcal{C}_{\text{Frame}}(\uparrow)$ | $\mathcal{E}_{\text{Warp}}(\downarrow)$ $(\times 10^{-3})$ | Agg. Score ($\downarrow$) |
| **VidEdit (ours)** | 28.1 (±3.0) | **91.5** (±11.1) | **21.7** (±8.4) | **3.06** | **0.077** (±0.054) | **0.730** (±0.109) | 22.6 (±3.6) | **3.01** | **97.4** (±1.4) | 5.2 (±9.3) | 2.33 |
| Text2Live | **28.7** (±2.8) | **94.1** (±14.6) | 20.4 (±6.0) | 3.07 | 0.155 (±0.035) | 0.710 (±0.088) | **22.8** (±2.9) | 4.04 | 97.0 (±1.4) | **3.9** (±4.9) | **2.00** |
| ControlNet | 28.0 (±2.6) | 84.8 (±24.0) | 18.7 (±6.6) | 3.30 | 0.647 (±0.061) | 0.312 (±0.036) | 10.8 (±1.5) | 12.85 | 86.2 (±3.6) | 177.9 (±64.3) | 46.74 |
| SDEdit | 26.1 (±2.9) | 65.7 (±31.8) | 14.2 (±7.4) | 3.96 | 0.490 (±0.051) | 0.377 (±0.034) | 17.9 (±1.6) | 9.57 | 83.9 (±5.2) | 71.2 (±30.1) | 19.42 |
| TAV | 27.5 (±3.1) | 73.4 (±36.3) | 13.7 (±9.0) | 3.92 | 0.584 (±0.079) | 0.274 (±0.060) | 13.0 (±2.1) | 12.00 | 96.4 (±1.6) | 47.8 (±27.7) | 13.27 |
| Pix2Video | **29.0** (±3.0) | 82.9 (±30.0) | 16.2 (±9.2) | 3.47 | 0.540 (±0.079) | 0.326 (±0.069) | 13.8 (±2.1) | 10.90 | 94.4 (±2.1) | 160.9 (±98.5) | 42.29 |

Even though our method is close to Text2Live in terms of frame accuracy and prompt consistency, the latter explicitly optimizes a generator on a CLIP-based loss, making the aforementioned metrics not reliable to assess its generalization performance and editing quality, as will be shown in the qualitative results. When it comes to image preservation evaluated with our similarity metrics, VIDEDIT outperforms all baselines in LPIPS and HaarPSI, and is similar to Text2Live on PSNR. This shows the capacity of our approach to optimally preserve the visual content of the source video while generating faithful edits to the target queries. Finally, VIDEDIT outperforms all methods in $\mathcal{C}_{\text{Frame}}$, and shows that the fine spatial control of our approach also translates in an improved temporal consistency.

To further analyze the fine-grained editing capacity of our method while preserving the original video content, we display for all baselines in Fig. 4, their local $\mathcal{A}_{\text{Frame}}$ score computed within a ground truth mask compared to an outer LPIPS metric (denoted O-LPIPS) computed on the invert of the mask. We see that VIDEDIT reaches a very good local frame accuracy, even outperforming Text2Live. Moreover, VIDEDIT shows a huge improvement on the O-LPIPS metric compared to the baselines, including Text2Live (3 vs 8), showing clearly a better preservation of out-of-interest regions.

Additionally, when comparing the processing time of different baselines, we found that VIDEDIT has a significant advantage, with a ∼ 30-fold speed-up factor over Text2Live. Focusing on the interative part of editing[2] in which users are interested in[3], Figure 5 underlines the lightweight aspect of our method. The panel on the left shows that VIDEDIT can perform a large number of edits on a 70 frame long video in significantly less time than other appoaches. As an illustration, VIDEDIT demonstrates approximately 30 times faster editing capabilities compared to Text2Live, which is the second leading baseline in terms of editing capacities. On the other hand, as depicted in the right panel, the use of VIDEDIT becomes increasingly time-efficient compared to other baselines, as the number of frames to edit growths.

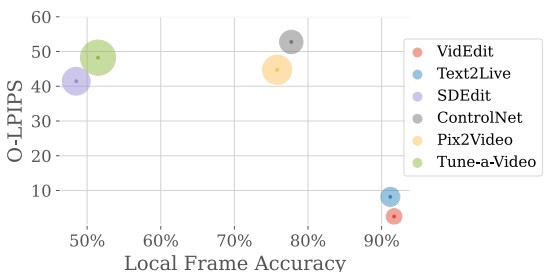

Figure 4: **Masked LPIPS vs Local Object Accuracy.** The size of each dot is proportional to the standard deviation of the local object accuracy.

**Qualitative results.** We show in Fig. 6 a visual comparison against the baselines to qualitatively assess the improvement brought out by our method.We can see that VIDEDIT performs fine-grained editing while perfectly preserving out-of-interest regions. In comparison to other baselines, the edits generated are more visually appealing and realistic. For example, VIDEDIT obtains a frame accuracy ($\mathcal{A}_{\text{Frame}}$) and prompt consistency ($\mathcal{C}_{\text{Prompt}}$) scores of 26.5 and 30 respectively compared to Text2Live which reaches 26 and 35 respectively. However, we can see that Text2Live's scores do not automatically translate into high-quality edits it often struggles to render detailed textures precisely localized on targeted regions. For example, ice creams are poorly rendered and some untargeted areas are being altered. Regarding Tune-a-Video and Pix2Video baselines, the methods are unable to generate a faithful edit at the exact location and completely degrade the

---

[2]Steps as DDIM inversion or LNA construction being considered as pre-processing steps.
[3]Editing the atlas and reconstructing the video for atlas based methods. Simply inferring the model for other baselines.

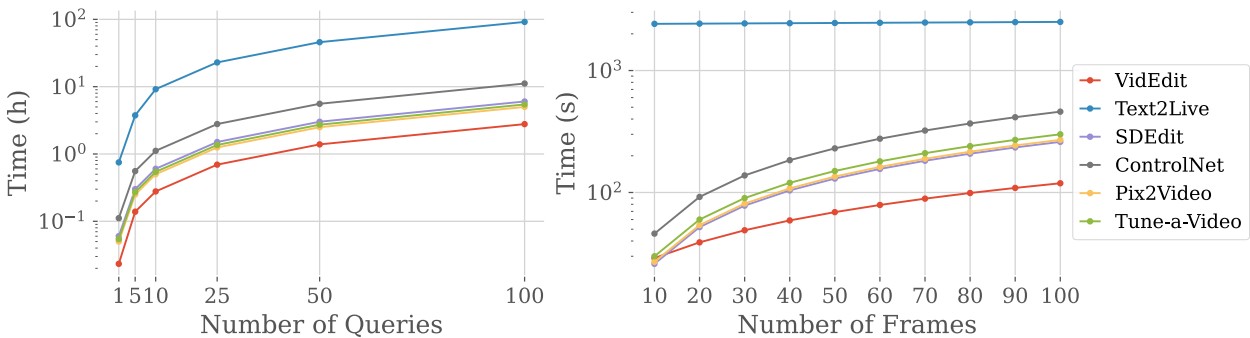

Figure 5: **Editing time.** VIDEDIT can edit videos significantly faster than existing methods.

**Source Prompt**: *"A couple of people riding a motorcycle down a road"*

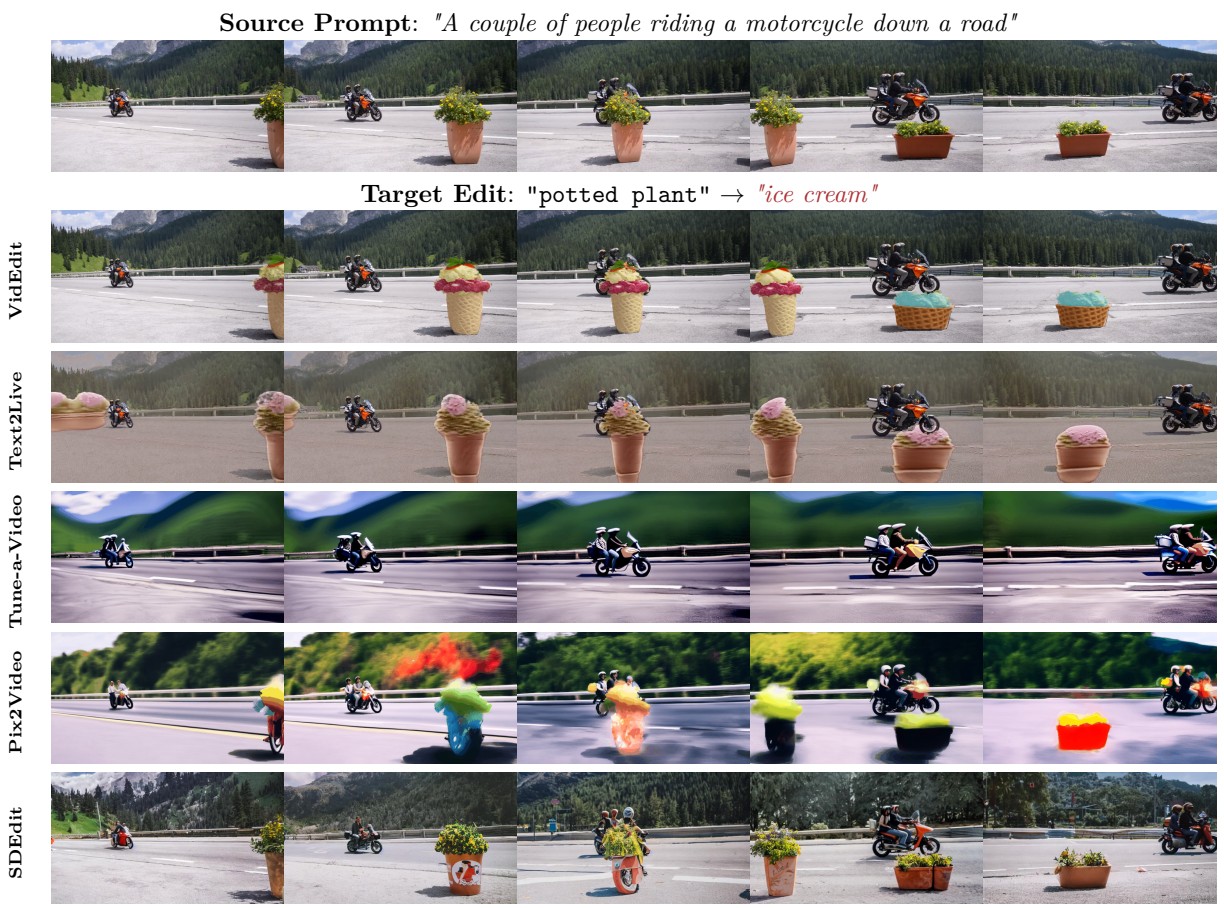

Figure 6: **Qualitative comparison of VidEdit with other baselines.**VIDEDIT generates higher quality textures than Text2Live. Tune-a-Video and Pix2video completely alters untargeted regions.

original content. Despite relatively high frame consistency scores in this video (96% for Tune-a-Video and 89% for Pix2Video *vs* 97.5% for VIDEDIT), noticeable flickering artifacts undermine the video content. On the other hand, naive frame-wise application of image-to-image translation methods also leads to temporally inconsistent results. For example, SDEdit is unable to both generate a faithful edit and to preserve the original content as it inherently faces a trade-off between the two. Other visual comparisons are shown in Appendix B.

### 4.3 Model Analysis

**Ablations.** We perform ablation studies to demonstrate the importance of our conditional controls once we map the edits back to the original image space. Tab. 2 compares the performance of our editing pipeline with both instance mask segmentation and HED edge conditioning against scenarios where these controls are disabled.

Table 2: **Ablation study.** Mask and HED map help to generate meaningful edits in the original frame space.

| Controls | | Semantic metrics | | | Similarity metrics | | |
|---|---|---|---|---|---|---|---|
| **Mask** | **HED** | $\mathcal{C}_{\text{Prompt}}$ ($\uparrow$) | $\mathcal{A}_{\text{Frame}}$ ($\uparrow$) | $\mathcal{S}_{\text{Dir}}$ ($\uparrow$) | LPIPS ($\downarrow$) | HaarPSI ($\uparrow$) | PSNR ($\uparrow$) |
| ✓ | ✓ | **28.1** (±3.0) | **91.5** (±11.1) | **21.7** (±8.4) | **0.077** (±0.054) | **0.730** (+0.109) | **22.6** (±3.6) |
| ✗ | ✗ | 25.5 (±3.1) | 64.3 (±38.3) | 10.6 (±7.5) | 0.099 (±0.051) | 0.632 (±0.131) | 20.1 (±4.0) |
| ✗ | ✓ | 26.3 (+3.0) | 72.4 (±34.0) | 13.0 (±7.6) | 0.095 (±0.049) | 0.672 (±0.110) | 20.8 (±3.6) |
| ✓ | ✗ | 27.5 (+2.8) | 81.9 (±24.2) | 18.0 (±8.4) | 0.081 (±0.042) | 0.639 (±0.128) | 20.7 (±3.3) |

In the case where no conditional control is passed on to the model, we observe a substantial drop in semantic metrics as the model generates edits at random locations in the atlas whose shapes don't match the structure of the target object. The introduction of edge conditioning without spatial awareness is quite similar to the previous case with the difference that the model tries to locally match the control information. This results in slightly better semantic results and similarity metrics than with no edge control. Finally, blending an edit with mask control without taking structure conditioning into account generates an edit at the right location but that is semantically incoherent once mapped back to the original images. Yet, this scenario achieves a decent prompt consistency as the objects still correspond to the target text query. We provide a visual illustration of this ablation study in Appendix A.

**Impact of hyperparameters.** We analyze in Fig. 7 VIDEDIT's behaviour versus the HED conditioning strength and noising ratio, respectively $\lambda$ and $\rho$. To analyze the trade-off between semantic editing and source image preservation, we compute a local LPIPS computed within a ground-truth mask, provided by DAVIS, versus a local CLIP score computed within the same mask for an edited object.

On the left panel, we can see that for $\lambda$ values lower than 0.4, the edge conditioning is not strong enough to guide the edits toward a plausible output on the video frames. This phenomenon is illustrated in Appendix A. On the contrary, for strength values larger than 1.2, the conditioning weighs too much on the model and hinders its ability to generate faithful edits. As expected, we notice that the local LPIPS decreases as the edge conditioning increases. While the decreasing rate is substantial between 0 and 1, the marginal gain diminishes for larger values.

Overall, setting the HED strength between 0.8 and 1.2 robustly enables to both perform faithful edits and preserve the original content. On the right panel, we see that both local CLIP score and LPIPS increase with the noising ratio. Indeed, for a null $\rho$ value, the region is reconstructed from the atlas, nearly identically to the original, and is then rewarded a low LPIPS. However, as no modification has been performed, the patch does not match the target text query and gets a lower CLIP score. As the noising ratio increases, the region de-

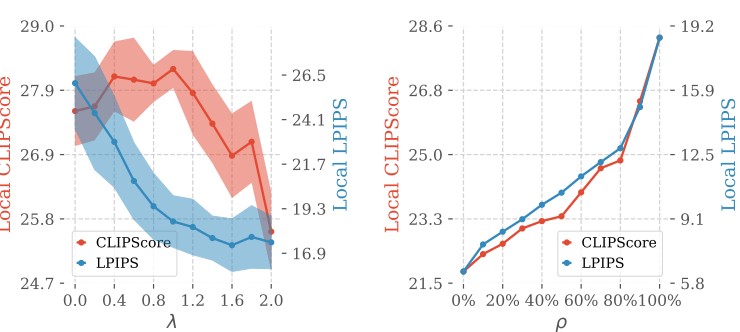

Figure 7: **VidEdit behavior wrt. different $\lambda$ and $\rho$ values.**

viates more from the input but also better matches the target edit. Note that for a $\rho$ value of 100%, the local LPIPS is constrained below 19, which still indicates a low disparity with the original image.

**Diversity.** Finally, we illustrate in Fig. 8 the capacity of VIDEDIT to produce various and sundry video edits from a unique pair *(video; target text query)*. In contrast, the randomness in Text2Live's training process only comes from the generator's weights initialization. As a result, method converges towards a unique solution and thus shows poor diversity in the generated samples.

**Target edit**: `"bus"` → *"a retrowave bus"*

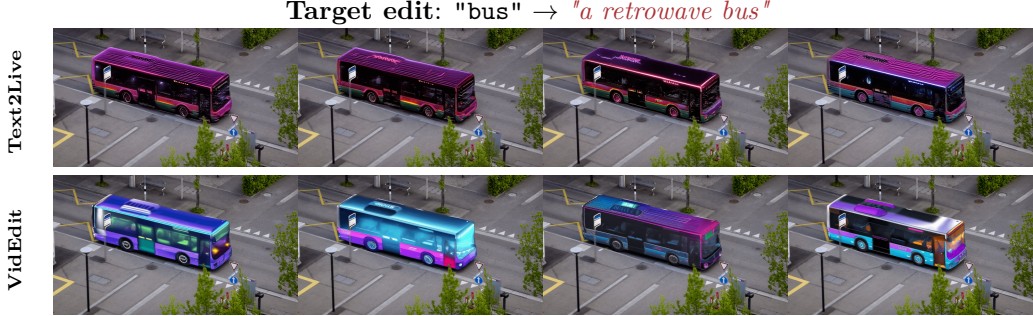

Figure 8: **Texture diversity.** We edit each video four times with the same input text query. Compared to Text2Live, our method is able to synthesize more diverse samples in much less time.

## 5 Conclusion & Discussion

We introduced VIDEDIT, a lightweight algorithm for zero-shot semantic video editing based on latent diffusion models. We have shown experimentally that this approach conserves more appearance information from the input video than other diffusion-based methods, leading to lighter edits. Nevertheless, the approach has a few limitations. Common with Kasten et al. (2021), the capacity of the MLP mapping networks decreases for complex videos involving rapid movements and very long-term videos. Since our method relies on the quality of such atlas representations, one possible way to expand the scope of possible video edits would be to strengthen and robustify the neural layered atlases construction approach.

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
