# A   Ablation Visualization

Fig. 9 illustrates the ablation study we led in Tab. 2. When VIDEDIT receives both conditional controls, it produces high quality results. Conversely, when these controls are deactivated, the model is free to perform edits at random locations in the atlas, resulting in uninterpretable visual outcomes. Enabling only the edge conditioning yields similar results, with the difference that the model attempts to locally match inner and outer edges. Finally, the sole use of a mask allows to perform edits at the correct locations, but that are semantically absurd once mapped back to the image space.

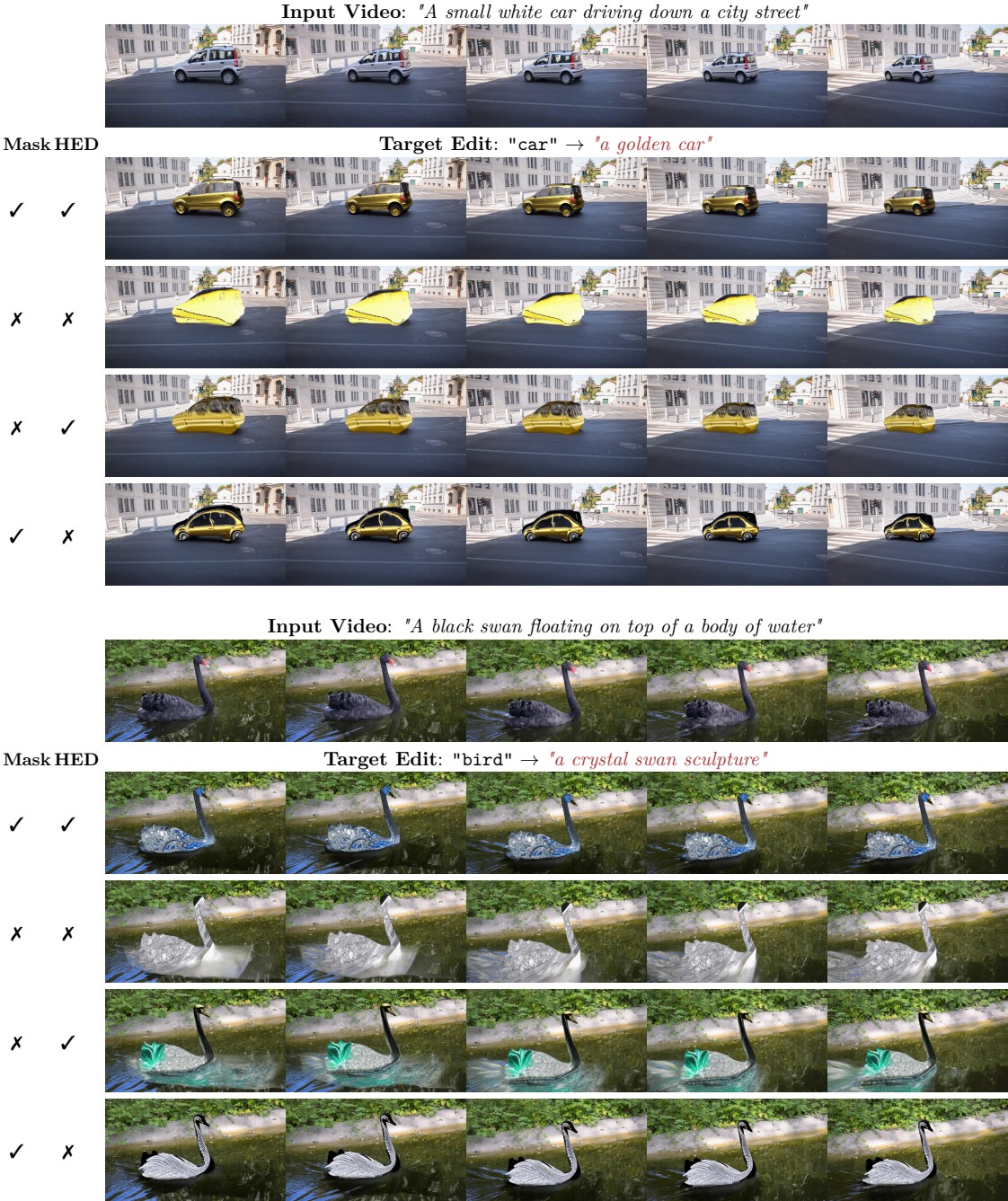

Figure 9: **Ablation visualization.**

# B    Additional Results

## B.1    VidEdit samples

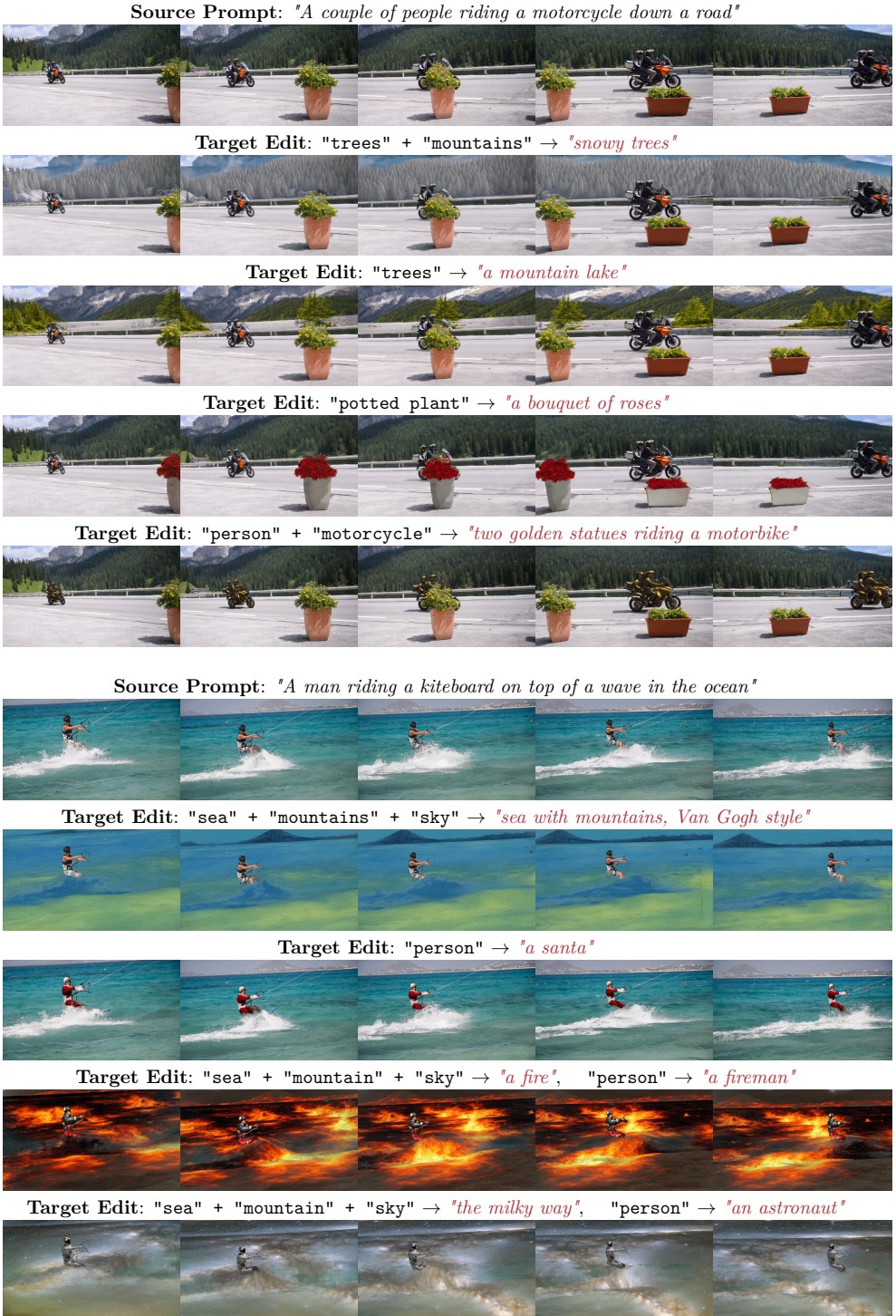

Figure 10: **Additional VidEdit sample results.**

## B.2 Baselines Comparison

Fig. 11 shows additional baselines comparison examples. We can see on both videos that VIDEDIT renders more realistic and higher quality textures than other methods while perfectly preserving the original content outside the regions of interest. The flamingo has subtle grooves on its body that imitate feathers and a fine light effect enhances the edit's grain. On the contrary, Text2Live struggles to render a detailed plastic appearance. The generated wooden boat also looks less natural and more tarnished than VIDEDIT's. Tune-a-Video and Pix2Video render unconvincing edits and completely alters the original content.

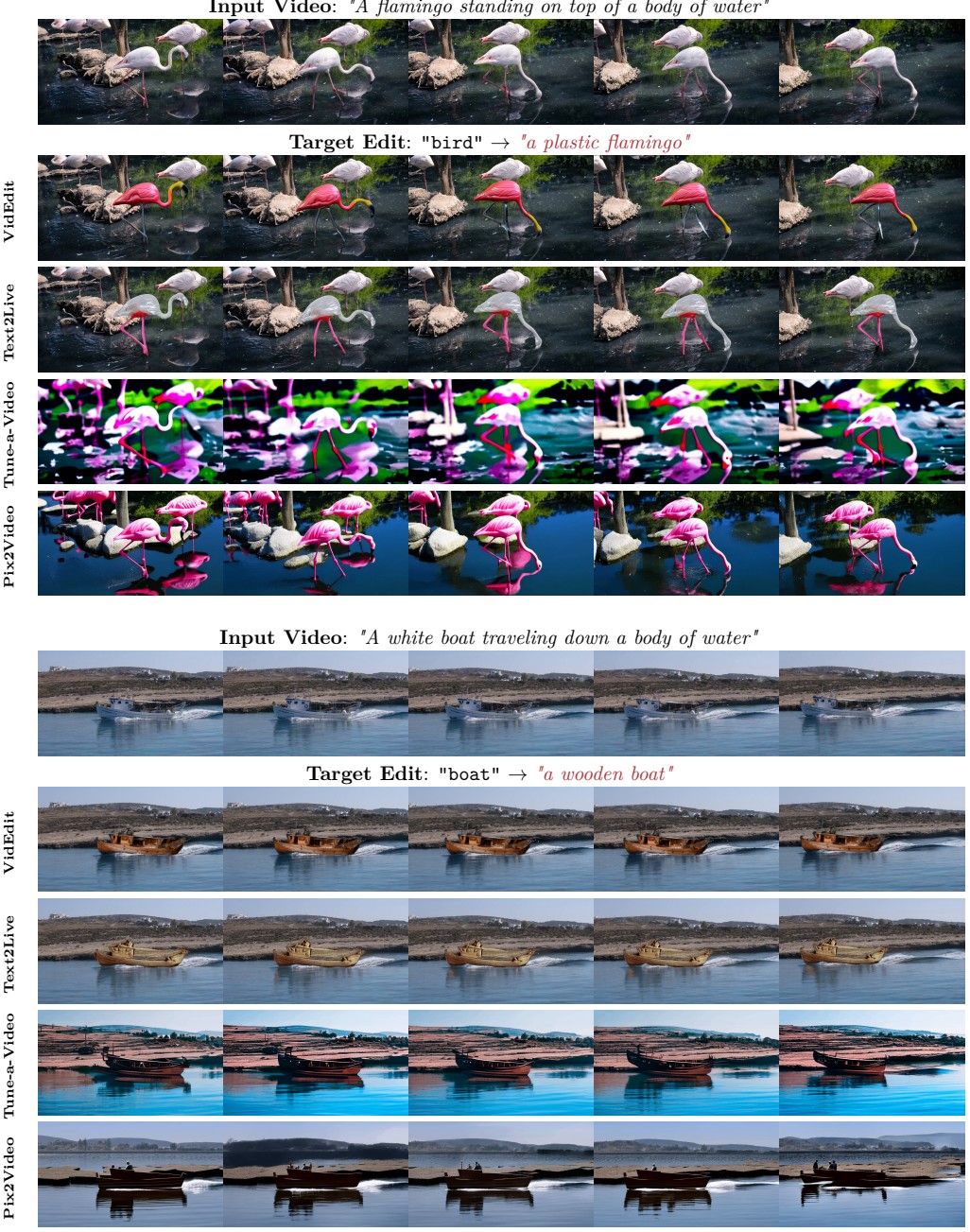

Figure 11: **Additional qualitative comparison between baselines.**

## C    Blending Effect

Fig. 12 shows the blending step's importance in the editing pipeline (Fig. 3). When considering only the RGB channels of a foreground atlas to infer an object's mask, the segmentation network has to deal with low contrasts between the background and the object, as well as duplicated representations within the overall atlas representation. This might lead to partially detected objects or masks placed at an incorrect location. In order to avoid these pitfalls, we leverage the atlas' alpha channel which indicates which pixels contain relevant information and must thus be visible. Therefore, we choose to blend the RGB channels with a fully white image according to the alpha values:

$$\mathbb{A}_{\text{Blended}} = \mathbb{A}_{\text{RGB}} \odot \alpha + \mathbb{I} \odot (1 - \alpha)$$

with $\mathbb{A}_{\text{RGB}}$ the RGB channels of an atlas representation, $\mathbb{I}$ a fully white image and $\alpha$ the atlas' opacity values.

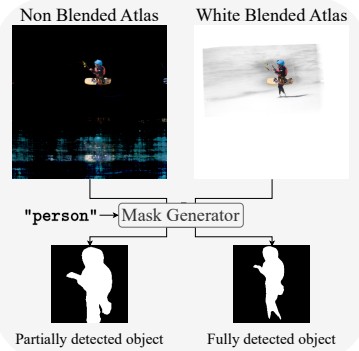

Figure 12: **Alpha blending effect**

## D    Atlas Construction

The atlas construction method takes as input a video and rough masks delineating the object(s) of interest. The objective is to compute (1) a collection of 2D atlases, one for the background and one for each dynamic object of interest; (2) a mapping from each pixel in the video to a 2D coordinate in each atlas; (3) opacity values at each pixel concerning each atlas. Each component is represented via coordinate-based MLPs. For the editing purpose, atlases are discretized into a fixed image grid ($1000 \times 1000$).

First, the mapping networks $\mathbb{M}_b, \mathbb{M}_f$ receive a pixel location $p = (x, y, t) \in \mathbb{R}^3$ as input and output its corresponding 2D point $(u, v) \in \mathbb{R}^2$ in each atlas

$$\mathbb{M}_b(p) = (u_b^p, v_b^p), \qquad \mathbb{M}_f(p) = (u_f^p, v_f^p)$$

The predicted 2D coordinates are then fed to an atlas network $\mathbb{A}$, that outputs the atlas' RGB color at that location. While separate networks $\mathbb{A}_f, \mathbb{A}_b$ could be learned to represent foreground and background, it is sufficient to use a single atlas $\mathbb{A}$, and restrict mapping networks $\mathbb{M}_b, \mathbb{M}_f$ to point into separate pre-defined quadrants in continuous [-1,1] space. The 2D atlas coordinates are then passed through a positional encoding denoted by $\phi(\cdot)$, to represent high frequency appearance information. The predicted colors are provided by:

$$\mathbb{A}(\phi(u_b^p), \phi(v_b^p)) = c_b^p, \qquad \mathbb{A}(\phi(u_f^p), \phi(v_f^p)) = c_f^p$$

In addition, each pixel location is also fed into the alpha MLP, $\mathbb{M}_\alpha(\phi(p)) = \alpha^p$ which outputs the opacity of each atlas at that location. The decomposition of the foreground and background layers is achieved by bootstrapping the alpha network using rough object masks that are computed with a pre-trained segmentor.

Utilizing these networks, the reconstructed RGB color at each video pixel is estimated by alpha-blending the corresponding atlas colors such that

$$c^p = (1 - \alpha^p)c_b^p + \alpha^p c_f^p$$

This framework is trained end-to-end, in a self-supervised manner, where the main loss is a reconstruction loss to the original video. Additionally, regularization losses on the mapping and decomposition enforce the learning of a meaningful and semantic atlas that can be used for editing:

1. **Rigidity loss**: The local structure of objects is preserved as they appear in the input video by encouraging the mapping from video pixels to atlas to be locally rigid.

2. **Consistency loss**: Corresponding pixels in consecutive frames of the video are forced to be mapped to the same atlas point; pixel correspondence is computed using an off-the-shelf optical flow method.

3. **Sparsity loss**: Mapping networks are encouraged to recover the minimal content needed to recover the video in atlases via a sparsity loss.

The total loss is given by:

$$\mathcal{L} = \mathcal{L}_{color} + \mathcal{L}_{rigid} + \mathcal{L}_{flow} + \mathcal{L}_{sparsity}$$

Additional details about these loss terms can be found in (Kasten et al., 2021). We follow the implementation setup described in this paper to obtain the discretized atlases that are used to perform editing.