# OpenReview forum: "VidEdit: Zero-Shot and Spatially Aware Text-Driven Video Editing"
_TMLR — Accepted by TMLR_

### Review · Reviewer_n9p4 · 2023-12-20

**Summary Of Contributions:**

The paper describes a method for text-based editing of videos. The proposed method uses the existing neural layer atlas model to decompose a video. A specific atlas can then be edited with the proposed editing diffusion pipeline which is the main contribution of the paper. A pretrained text-to-image diffusion model is used and additionally conditioned based on the extracted segmentation mask and edge detection. The way how the atlas is edited distinguishes the method from the previous work. It is simple and fast and does not require any additional training. The resulting images are diverse and visually convincing.

**Audience:**

Yes

**Broader Impact Concerns:**

is already addressed in the paper

**Claims And Evidence:**

No

**Requested Changes:**

- The main weakness of the paper is the evaluation. The evaluation metrics, especially the temporal consistency needs further explanation and justification. Further, visual results as a video would be required to assess the perceptual and temporal quality and ideally supplemented with a user study.
- Further, the overall presentation of the paper can be improved (see comments/suggestions above).

**Strengths And Weaknesses:**

**Strengths**

-	Simple method using existing pretrained modules in an effective way. No further training needed.
-	Does not require training at test-time and therefore allows for very fast inference.
-	The visual results are convincing (video is missing). Method is able to generate diverse edits.
-	The proposed conditioning is highly effective as the ablation shows.
-	Addresses the challenge of evaluating editing with some proposals of quantitative metrics, however, these metrics could be better motivated and evaluated that they measure the right thing (see below).

**Weaknesses**

*Evaluation metrics:* I appreciate the attempt to provide some quantitative measurements, however, I am not convinced by their suitability:

-	The measurement for temporal consistency is calculated by the clip similarity between two frames. Besides that it is unclear, whether it uses the image or textual embedding, it is also unclear to me, how the similarity of a single clip score per frame corresponds to spatial temporal stability of a video. That seems a very coarse measurement to me and more evidence needs to be provided that this novel metric can capture temporal consistency. Alternatives on using e.g., warping errors are more detailed. Better would be also to attach a video and user study, as qualitative analysis is still very important to evaluate perceptually, whether a video is temporal stable. Given that obtaining strong temporal consistency is one of the main claims of the paper I would expect more evidence for that.
-	Content preservation is measured as the similarity between the edited and original. However, in the main description it is unclear whether this is applied to the full image or weighted by the editing mask M, otherwise the best results can be obtained with no editing at all and therefore can be misleading.

*Lacks clarity:*

-	How is the atlas to be edited chosen?
-	Will the code be made available?

*Improvements in presentation:*

-	Not all variables are introduced/defined, e.g., t, alpha in Eq. 1, M in Eq. 2, variables in the frame consistency measurement, …
-	Figures and Captions are missing references of used methods, e.g., in Fig. 2 reference to NLA models by Kasten et al. (2021), or used mask generator, HED in Fig. 3.
-	Adding all equation numbers, makes referencing them possible.

---

> ### Author Response · Authors · 2024-02-02
>
> We appreciate the reviewer's feedback and the opportunity to clarify our contributions.
>
> * **The measurement for temporal consistency is calculated by the clip similarity between two frames. Besides that it is unclear, whether it uses the image or textual embedding, it is also unclear to me, how the similarity of a single clip score per frame corresponds to spatial temporal stability of a video**
>
> As in [1] and [2] we evaluate the temporal consistency of an edited video with the CLIP similarity between two consecutive frames. For each pair of consecutive frames in a video, we compute their CLIP image embeddings and measure their proximity with a cosine similarity. A highly temporally consistent edited video will produce consecutive frames that are close together in the CLIP embedding space and thus have a high similarity score.
>
> * **Given that obtaining strong temporal consistency is one of the main claims of the paper I would expect more evidence for that.**
>
> The CLIP similarity metric between consecutive frames is a proxy to estimate temporal consistency but we agree that this could be a coarse measurement. As suggested, in order to provide additional evidence of the temporal smoothness of our edited videos, we evaluated the warping error of our edited videos. Unlike our metric based on CLIP, which captures overall semantic changes between consecutive frames, the warping error focuses on capturing specific, localized spatial changes.
>
> |            | Warp Error (x10^-3) |
> |:----------:|:-------------------:|
> |   VidEdit  |         5.2         |
> |  Text2Live |         3.9         |
> | ControlNet |        177.9        |
> |   SDEdit   |         71.2        |
> |     TAV    |         47.8        |
> |  Pix2Video |        160.9        |
>
> The reported metric in Table 1. is the Mean Squared Error between an original edited frame and the predicted/warped edited frame. Table 1. suggests that our method effectively provides a strong temporal consistency, that is on par with Text2Live that optimizes an edit layer for each provided editing text prompt.
>
> * **Better would be also to attach a video and user study, as qualitative analysis is still very important to evaluate perceptually, whether a video is temporal stable.**
>
> To better assess the visual quality of our edited videos, we provide a link to visualize some of our editing results: https://anonymous.4open.science/r/videdit-videos-1BDC/VidEdit-samples/
>
> For example, in the “wooden-swan” video, our approach transforms the swan into a wooden sculpture with sharp details and texture and interplay of lights. The rest of the video content remains nearly identical to the original. In contrast, baseline methods such as Tune-a-Video or Pix2Video not only alter the bird but also significantly modify the entire resulting in a poor match with the original. Moreover, these methods lack temporal smoothness. As for Text2Live, the textures and details for the wooden swan sculpture are not convincing.
>
> * **Given that obtaining strong temporal consistency is one of the main claims of the paper I would expect more evidence for that.**
>
> We do not claim to reach a better temporal consistency than Text2Live, but we propose an efficient method that allows for very precise local and global editing (see Table 1, Figure 4, Figure 6 and Supplementary B)  with high temporal coherence (see Table 1, and provided videos) , that is empirically on par with Text2Live in terms of temporal smoothness.
>
> * **However, in the main description it is unclear whether this is applied to the full image or weighted by the editing mask M, otherwise the best results can be obtained with no editing at all and therefore can be misleading.**
>
> In Table 1. content preservation metrics are computed over the full video frames and not weighted by the editing mask M. The similarity metrics shown in Table 1 suggest a high level of content preservation.  However, these metrics shall be interpreted in conjunction with the semantic metrics which demonstrate how well our edited videos align with the provided text prompts. If we do not edit the videos at all, our method would reach very high content preservation metrics at the expense of poor semantic scores.
>
> In addition, in Figure 4, we performed a more detailed experiment illustrating our method's editing capabilities while maintaining the integrity of the original video content. We calculate a local semantic score  within a ground truth mask and compare it with a content preservation metric computed outside this mask. The perfect method would reach a 100% frame accuracy within the mask while perfectly preserving the video content outside this mask (i.e O-LPIPS = 0). Our results demonstrate that our method achieves the highest local frame accuracy in relation to Outer-LPIPS when compared to all the baseline methods, which is the desired capacity of our method.

---

> > ### Author Response · Authors · 2024-02-02
> >
> > * **How is the atlas to be edited chosen?**
> >
> > Our method is applicable on both foreground or background atlases. In our quantitative experiments, we choose to focus on foreground atlases.
> >
> > * **Will the code be made available?**
> >
> > Code will be made available .
> >
> > [1] Wu, Jay Zhangjie, et al. “Tune-A-Video: One-Shot Tuning of Image Diffusion Models for Text-to-Video Generation.” 2023 IEEE/CVF International Conference on Computer Vision (ICCV), IEEE, 2023. Crossref, https://doi.org/10.1109/iccv51070.2023.00701.
> >
> > [2] Ceylan, Duygu, et al. “Pix2Video: Video Editing Using Image Diffusion.” 2023 IEEE/CVF International Conference on Computer Vision (ICCV), IEEE, 2023. Crossref, https://doi.org/10.1109/iccv51070.2023.02121.

---

> ### Author Response · Authors · 2024-03-04
>
> Dear Reviewer,
>
> We reiterate our appreciation for your time. We think that your concerns can be addressed and respectfully ask you to read our response and check the changes we did in the draft. As you requested, we provided an additional metric (Warp Error) to highlight the temporal coherence of our edited videos as well as visual examples in the repository to further illustrate our qualitative results. If possible, we respectfully ask you to engage in discussion with us if you feel your concerns have not been addressed. We are hopeful that your time allows continual discussion so you can make your final recommendation when all your concerns are addressed.

---

### Review · Reviewer_2v7z · 2024-01-20

**Summary Of Contributions:**

This paper presents a zero-shot video editing method, combining the Neural Layered Atlas (NLA) approach with Stable Diffusion (SD) and an edge estimator and Mask2former panoptic segmentation. The idea is to reproduce the NLA approach, and then replace the manual editing of atlases with DDIM-based editing, where the DDIM is aided slightly by the edge map, and guided by a language prompt via classifier-free guidance.

**Audience:**

Yes

**Claims And Evidence:**

No

**Requested Changes:**

Overall, I would like to see the technical details clarified, specifically with respect to the implementation choices in the atlas/layering parts (especially highlighting differences from the original work), and also clarifying the modifications to SD. Beyond these major issues, I have a long list of minor issues I uncovered while reading.


"VidEdit modifies the original content by precisely delineating the regions of interest" This claim should be clarified. It sounds like the modification is to delineate the region, but I think this is not what the method actually achieves.

The paper says that "spatially grounded" editing "is achieved by extracting a crop around the area of interest and intentionally utilizing a non-invertible noising process." What is meant here? I can't imagine something like this being implemented unintentionally, but maybe I am not reading this right.

The paper claims that diffusion-based models, unlike GANs, "can be reliably trained on massive amounts of data and produce convincing samples". I think this is basically a false claim. GANs can also be trained on massive amounts of data, and they can also produce convincing samples.
The paper claims, in 3.1, that "The 2D atlases obtained by disentangling the video are a well-posed framework to edit objects while ensuring a strong temporal consistency." I am not convinced about this. In general this seems like an empirical claim that needs evidence, but also I am not sure what the authors mean by "well-posed framework".

In the related work, it would be great to connect the works to the current effort somehow. What are the specific similarities and differences, with respect to the current work?

"these methods still suffer from annoying flickering artifacts" It is not nice to say that the outputs of other methods are "annoying" to you. About Text2Live, the related work section reports that "its training procedure for each prompt makes it impractical to use". I checked that paper and it does not look impractical -- it's a zero-shot method, and it's an ECCV 2022 oral, and it uses a similar layered "atlas" representation as this work.


"The ergonomy and performance of text-conditioned diffusion models are a great motivation to perform those editions based on text queries." I am not sure about many parts of this sentence. Ergonomy? Perform which editions, and editions of what?


"the automatic video editing charge" -- What is this? I think this is only mentioned once, but it seems important.


"First, the inverting procedure is relatively time-consuming. We thus opt to limit the computational overhead in order to increase our editing efficiency." I do not follow this. Why is the computation-saving motivated by a time-consuming procedure?


"we choose not to perform inverse DDIM in the latent space to encode x0 for two main reasons." As far as I can tell, this is the first occurrence of x0. What is x0?

The paper mentions that the approach leads to "lighter edits" than prior work, but I am not sure this is a good thing.

The paper says "the capacity of the MLP mapping networks decreases for complex videos involving rapid movements and very long-term videos". Why does the capacity decrease? Capacity seems like something you can control, and for longer videos, you should probably increase it instead of decrease it. Perhaps the authors can clarity this strategy.

The paper claims that "one possible way to expand the scope of possible video edits would be to strengthen and robustify the neural layered atlases construction approach." I do not know what this means. How would the strengthening be done exactly, and what robustness is being referred to, and how would these things affect the scope of the possible edits?

The paper says that they propose a model "which by design fulfills temporal smoothness", but I think the temporal smoothness comes from the atlas component, which is not a contribution of this paper really.

The paper concludes with a quote from the Stable Diffusion paper, about ethical and legal use of stable diffusion. I do not think this is a useful conclusion to the paper.

**Strengths And Weaknesses:**

I think this is a reasonable combination of components, perhaps with exception of the edge map estimator from HED, which might be redundant since the model has access to panoptic segmentation from Mask2former.

I am not sure that the method, as proposed, is complete. Using the M_f and M_b networks, we can apparently map the video's xyt coordinates onto uv coordinates, and these uv coordinates can be fed to an A network, and an alpha map for blending, but there is no learning described here -- no optimization, no reconstruction, no actual learning of the M and A networks. To learn about these steps, we need to read a different paper, Kasten et al 2021. Looking into that paper, it seems like maybe two A networks are necessary (but not mentioned here) -- is it A_f and A_b, or just "A" as described? What losses are used here? Is it color+rigidity+consistency+sparsity like the referenced paper? How does the panoptic segmentation fit in here? In the original paper it was a bootstrapping process, which began with MaskRCNN.

The paper mentions modifying SD by "incorporating semantic guidance based on high-resolution semantic masks". How are the semantics incorporated here exactly? How exactly was the architecture modified? These seem like crucial method details, and yet the paper only provides a sentence or two.

A downside with the layering approach is that it assumes that the scenes are basically planar, with very little occlusion happening. It would be nice to resolve this with the diffusion somehow.

---

> ### Author Response · Authors · 2024-02-02
>
> We appreciate the reviewer’s feedback and the opportunity to clarify our contributions
>
> * **I think this is a reasonable combination of components, perhaps with exception of the edge map estimator from HED, which might be redundant since the model has access to panoptic segmentation from Mask2former.**
>
> While the semantic mask guarantees precise local object editing by restricting alterations to the target area, the HED edge map plays a crucial role in preserving the overall structure within the defined mask. Indeed, since the representations of foreground objects in the atlas space tend to be distorted, mainly due to their large motion, it is essential to maintain their semantic structure i.e inner edges, as closely as possible to the original, ensuring the preservation of the original motion. As demonstrated in Table 2 and Supplementary A, these two conditionings bring complementary information to perform fine-grained editing.
>
> * **I am not sure that the method, as proposed, is complete. [...] In the original paper it was a bootstrapping process, which began with MaskRCNN.**
>
> This paper introduces a video editing method based on video atlas representations. In our approach, we take for granted the creation of the atlas. Specifically for our methodology, we treat its training as a black-box, refraining from any participation in the details of the training process. After the training phase, we only utilize the networks $\mathbb{M}_b,\mathbb{M}_f, \mathbb{A}, \mathbb{M}{\alpha}$ to produce the visual representation of the atlas, which is then employed in our editing procedure. As we do not modify the atlas construction pipeline, we purposely omit some details and we refer to the original paper for additional information.
>
> * **The paper mentions modifying SD by "incorporating semantic guidance based on high-resolution semantic masks". How are the semantics incorporated here exactly? How exactly was the architecture modified? These seem like crucial method details, and yet the paper only provides a sentence or two.**
>
> To integrate our semantic information to our diffusion model, we leverage a ControlNet version of a Stable Diffusion Model [1]. As in the original Stable Diffusion, the text-prompt conditions the UNet via cross-attention layers at each block. For the edge conditioning, ControlNet has learned to detect and to integrate HED edges as conditional information to a diffusion model via training a copy of its pre-trained layers while also maintaining locked the pre-trained parameters separately. The trainable and locked copies of the parameters are then connected at each block of the UNet decoder via “zero convolution” layers that are also optimized. We refer to the original paper for additional information. When it comes to the segmentation mask, as depicted in Figure 2, we perform a classical Blended Latent Diffusion [2], [3], described in Equation 2, to edit our patch inside the mask while preserving the area outside.
>
> * **"VidEdit modifies the original content by precisely delineating the regions of interest" This claim should be clarified. It sounds like the modification is to delineate the region, but I think this is not what the method actually achieves.**
>
> In order to edit an object in the video, we aim to edit its representation in the atlas space. To localize and delineate the area to edit in this space, we make use of a pre-trained segmentation network. This is necessary in order to constraint our edit to the targeted object when we project the edit back to the video. We modify this sentence with: *"VidEdit precisely delineates the region of interest in the atlas space and modifies its content while leaving the remainder unchanged"*
>
> * **The paper says that "spatially grounded" editing "is achieved by extracting a crop around the area of interest and intentionally utilizing a non-invertible noising process." What is meant here? I can't imagine something like this being implemented unintentionally, but maybe I am not reading this right.**
>
> Contrary to DiffEdit that advocates to perform a DDIM inversion for subtle image editing, we purposely choose not to perform DDIM inversion but a regular noising process in order to allow fast inference and to offer the possibility to generate various edits based on a single text prompt. In practice, this holds significant importance as it allows the user to select the preferred edit. We qualitatively illustrate our greater sample diversity in Figure 8 where we compare 4 edits obtained by Text2Live and our method on a unique video with a single text prompt.

---

> > ### Author Response · Authors · 2024-02-02
> >
> > * **The paper claims that diffusion-based models, unlike GANs, "can be reliably trained on massive amounts of data and produce convincing samples". I think this is basically a false claim. GANs can also be trained on massive amounts of data, and they can also produce convincing samples.**
> >
> > We respectfully disagree with the reviewer. GANs are notoriously difficult to train due to their adversarial training objective and suffer from various pitfalls that can hinder their ability to produce diverse samples. To the best of our knowledge, GigaGAN [4] , which stands as the state-of-the-art GAN, was trained on a notably smaller dataset compared to the state-of-the-art diffusion models and does not surpass them in terms of image quality. However, we are willing to temper this statement with:
> >
> > *“In contrast to Generative Adversarial Networks which are notoriously difficult to train, diffusion models offer a more reliable training process and consistently generate highly convincing samples.”*
> >
> > * **The paper claims, in 3.1, that "The 2D atlases obtained by disentangling the video are a well-posed framework to edit objects while ensuring a strong temporal consistency." I am not convinced about this. In general this seems like an empirical claim that needs evidence, but also I am not sure what the authors mean by "well-posed framework".**
> >
> > The underlying assumption under the atlas construction method for video editing, shared by NLA and  Text2Live, is that a video content can effectively be projected in a 2D space where objects will have minimal distortion with respect to their real-world appearance. Under this assumption, we demonstrate in this paper that atlas representations form a well-defined framework to perform a video editing task, treating it as an diffusion-based image editing task, which allows to efficiently edit a video and grant flexible user control with the joint combination of provided conditional information.
> >
> > * **In the related work, it would be great to connect the works to the current effort somehow. What are the specific similarities and differences, with respect to the current work?**
> >
> > We rewrote the **Text-driven Video Editing** paragraph in the related work section to better highlight our similarities and differences with existing works. Specifically, we divide video editing methods into two categories: diffusion based methods that adapt a pre-trained diffusion model to video editing and altas based methods that decompose a video in a set of 2D atlases that can be manipulated to edit videos.
> >
> > * **"these methods still suffer from annoying flickering artifacts" It is not nice to say that the outputs of other methods are "annoying" to you.**
> >
> > We do not aim to diminish in any way the impact or the scope of other methods. We agree the term “annoying” might be inappropriate to describe concurrent works. Hence we would like to reformulate with: “However, these methods still exhibit flickering artifacts and inconsistencies that alter the visual quality of edited contents.”
> >
> > * **About Text2Live, the related work section reports that "its training procedure for each prompt makes it impractical to use". I checked that paper and it does not look impractical -- it's a zero-shot method, and it's an ECCV 2022 oral, and it uses a similar layered "atlas" representation as this work.**
> >
> > Once again, we do not aim to diminish the work from the authors of Text2Live. In practice Text2Live requires a costly optimization over an atlas layer for editing. As per the original paper's implementation, this optimization procedure can extend up to 40 minutes (with our NVIDIA RTX) , a duration we consider prohibitive to perform multiple edits on a single video. In contrast we propose a method that can edit a video in about 1 minute, which de facto implies a greater range of application than Text2Live.  However, we are willing to modify this sentence as:
> >
> > *“Based on neural layered atlases, Text2Live allows coherent text-to-video (T2V) editing but its costly optimization for each prompt limits its ability to produce edits on the fly.”*
> >
> > * **The ergonomy and performance of text-conditioned diffusion models are a great motivation to perform those editions based on text queries." I am not sure about many parts of this sentence. Ergonomy? Perform which editions, and editions of what?**
> >
> > We modify this sentence with the following (Section 3: VidEdit Framework):
> >
> > *“The high visual quality offered by T2I diffusion models as well as their effectiveness to generate samples that are aligned with provided conditional information motivate us to utilize these models to perform our video editing task in the 2D atlas space.”*

---

> > > ### Author Response · Authors · 2024-02-02
> > >
> > > * **"the automatic video editing charge" -- What is this? I think this is only mentioned once, but it seems important.**
> > >
> > > “The automatic video editing charge” refers to the task of video editing.
> > > We have reformulated this sentence (Section 3.1: Zero-Shot Atlas Editing) into:
> > >
> > > *As illustrated with \cref{global_videdit}, the automatic video editing task is transformed into a much straightforward, training free, and adaptable image editing task, resulting in competitive performance.*
> > >
> > > * **"First, the inverting procedure is relatively time-consuming. We thus opt to limit the computational overhead in order to increase our editing efficiency." I do not follow this. Why is the computation-saving motivated by a time-consuming procedure?**
> > >
> > > DiffEdit provides theoretical justification that advocates for the use of a DDIM inversion procedure to obtain image latents in order to perform image editing. However, we do not need such an inversion process and choose not to use it in our method to reduce inference cost and to enable diverse editing. We rewrite this section to make it more understandable (Section 3.2 Step 2)
> > >
> > > * **"we choose not to perform inverse DDIM in the latent space to encode $x_0$ for two main reasons." [...] What is $x_0$?**
> > >
> > > $x_0$  is the image latent associated with the patch we aim to edit. We rewrite this section to define what $x_0$ refers to (see section 3.2: Step 2 Noising steps)
> > >
> > > * **The paper mentions that the approach leads to "lighter edits" than prior work, but I am not sure this is a good thing.**
> > >
> > > For foreground objects that have large motions, we constrain our editing via the HED map to preserve their semantic structure in the atlas space so that the motion is coherent once we project the edit back to the video space. For these objects, our method can generate video edits that correspond to a given text prompt with maximal structure preservation. (See Appendix A and B)
> > >
> > > * **The paper says "the capacity of the MLP mapping networks decreases for complex videos involving rapid movements and very long-term videos". [...] Perhaps the authors can clarify this strategy.**
> > >
> > > The longer the video, the more it is susceptible to contain complex motions from either the subjects or the camera and the harder it will be to represent the content in a 2D space. In order to adapt to such constraints, it would most likely need to rethink the architecture of the MLPs (using larger MLPs) and result in heavier training. Additionally, a possible way to adapt to longer videos, would be to learn multiple atlases to represent different sequences of the video and match the edits across the atlases with for example.
> > >
> > > * **The paper claims that "one possible way to expand the scope of possible video edits would be to strengthen and robustify the neural layered atlases construction approach."[...] how would these things affect the scope of the possible edits?**
> > >
> > > Layered Neural Atlases can sometimes fail to accurately represent the original content with minimal distortion in complex video with large object motion and camera motion. In this case, especially foreground objects might not have a meaningful representation on the 2D atlas which makes it difficult to edit. In order to alleviate this problem and enlarge the scope of possible edits, various strategies could be explored such as :
> > >
> > > * Modifying the architecture of the MLPs
> > >
> > > * Using additional scene information (depth for example…)
> > >
> > > * Separate a video into multiple atlases and match the edits from one atlas to another.
> > >
> > > Those are an intrinsic limitation of Atlas that are perspectives for future works.
> > >
> > > * **The paper says that they propose a model "which by design fulfills temporal smoothness",[...] which is not a contribution of this paper really.**
> > >
> > > Indeed, the temporal smoothness comes from the atlas component. We claim with this method that atlas based video editing can be treated as an image video editing task via a well conditioned pre-trained image diffusion model applied on a video atlas representation. The temporal smoothness is ensured by the atlas representation while the flexible user control is granted by the conditional information provided to the generative models.
> > >
> > > * **The paper concludes with a quote from the Stable Diffusion paper, about ethical and legal use of stable diffusion. I do not think this is a useful conclusion to the paper.**
> > >
> > > We remove this part from the conclusion.
> > >
> > > [2] Avrahami, Omri, et al. “Blended Latent Diffusion.” ACM Transactions on Graphics, vol. 42, no. 4, July 2023, pp. 1–11. Crossref, https://doi.org/10.1145/3592450.
> > >
> > >
> > > [3] Couairon, Guillaume, et al. "Diffedit: Diffusion-based semantic image editing with mask guidance." arXiv preprint arXiv:2210.11427 (2022).
> > >
> > > [4] Kang, Minguk, et al. “Scaling up GANs for Text-to-Image Synthesis.” 2023 IEEE/CVF Conference on Computer Vision and Pattern Recognition (CVPR), IEEE, 2023. Crossref, https://doi.org/10.1109/cvpr52729.2023.00976.

---

> > ### Comment · Reviewer_2v7z · 2024-02-22
> >
> > For the HED part: would it also work to just run a Canny edge detector (like from opencv) on the panoptic segmentation map?
> >
> > For the incomplete method: despite the argument that this is intentional, I still feel that the paper would be better if you described the missing parts of the method, so that the paper makes more sense on its own.
> >
> > About controlnet: I think I understand the summary written here, but it's not clear to me exactly how this fits into the paper, and again I am not sure why this component (which sounds important) is not really described in the main text. Right now it's just one line in "4.1 Experiment setup".
> >
> > About the delineation: again, this sort of information belongs in the main text. I see that the original sentence has been slightly rewritten, but new information has not really been added.
> >
> > About intentional vs. non-intentional non-invertible noising -- I still doubt that the prior work implemented anything unintentionally.

---

> > > ### Author Response · Authors · 2024-02-26
> > >
> > > * **For the HED part: would it also work to just run a Canny edge detector (like from opencv) on the panoptic segmentation map?**
> > >
> > > As for the HED map, a Canny edge map would also help to some extent to control the semantic layout of the edited area. However, an HED map offers several advantages over a Canny edge detector, especially at multi-scales and complex structures patterns handling that translate into better performance to control the semantic structure of the edit. Moreover, the edges produced by the HED detector align more closely with the boundaries of semantic objects, while a Canny detector may generate extraneous edges that overly restrict editing.
> > >
> > > * **For the incomplete method: despite the argument that this is intentional, I still feel that the paper would be better if you described the missing parts of the method, so that the paper makes more sense on its own.**
> > >
> > > To ensure comprehensive coverage, we include some details regarding the atlas construction method in the appendix section (Supplementary Section D. Atlas Construction). Nevertheless, since this method is not a contribution from our paper and merely serves as a starting point for our approach, we believe it is unnecessary to provide an exhaustive description of its construction process. We refer to the original NLA [1] paper for supplementary technical details.
> > >
> > > * **About controlnet: I think I understand the summary written here, but it's not clear to me exactly how this fits into the paper, and again I am not sure why this component (which sounds important) is not really described in the main text. Right now it's just one line in "4.1 Experiment setup".**
> > >
> > > ControlNet is the diffusion model we use to perform our denoising steps conditioned by the edge map and the text prompt. Its utilization is described in equation 3) where $\epsilon_{\theta}$ refers to the ControlNet U-Net. As we simply make use of a pre-trained model without modifying its architecture, specific details in its architecture are not essential and we did not deem important to include this part in the paper. However, we follow your recommendation and integrate some details in the paper (section 4. VidEdit Setup). Architectural details that provide an extensive understanding on how the conditioning modalities are incorporated can be found in the original paper. Finally, as the source code will be made available, comprehensive details will be provided to replicate the results.
> > >
> > > * **About the delineation: again, this sort of information belongs in the main text. I see that the original sentence has been slightly rewritten, but new information has not really been added.**
> > >
> > > Following your recommendation, we add additional information to this sentence:
> > >
> > > *Given an input video and a target edit,e.g. "road" → "a night sky" VidEdit precisely delineates the region of interest in the atlas space as well as the internal edges that characterize its semantic structure. The text prompt and the edge map are then passed to a pre-trained conditional diffusion model that generates an edit that matches these controls. During the generation phase, the edit seamlessly merges with the original atlas through a blended diffusion process, which leaves the remainder of the video content unchanged.*
> > >
> > > * **About intentional vs. non-intentional non-invertible noising -- I still doubt that the prior work implemented anything unintentionally.**
> > >
> > > There seems to be a misunderstanding. In this paper we **intentionally** opt for a non-invertible noising, in contrast to DiffEdit which **intentionally** uses an invertible noising method. We want to clarify that we are **not** asserting that previous research unintentionally implemented this choice. Our setup allows us to be able to generate various edits.
> > >
> > >
> > > [1] Kasten, Yoni, et al. “Layered Neural Atlases for Consistent Video Editing.” ACM Transactions on Graphics, vol. 40, no. 6, Dec. 2021, pp. 1–12. Crossref, https://doi.org/10.1145/3478513.3480546.

---

> > > > ### Comment · Reviewer_2v7z · 2024-02-26
> > > >
> > > > small note: I'm not saying to run Canny on the image, run it on the panoptic segmentation map.

---

> > > > > ### Author Response · Authors · 2024-02-26
> > > > >
> > > > > A Canny edge detector applied on top of the panoptic segmentation will only output a single outer contour for each region. In contrast, the HED detector applied on the input image extracts finer information based on inner contours of objects, which gives much more control on the edition.

---

### Review · Reviewer_xGev · 2024-01-22

**Summary Of Contributions:**

In this paper, the authors propose VidEdit, a novel method for zero-shot and spatially aware text-driven video editing. The authors aim to address the limitations of existing methods in video editing and introduce a new approach that combines atlas-based and pre-trained text-to-image diffusion models. By leveraging off-the-shelf panoptic segmenters and edge detectors, VidEdit enables precise spatial control over targeted regions while preserving the original video's structure. The experiments conducted on the DAVIS dataset demonstrate the superiority of VidEdit over state-of-the-art methods in terms of semantic faithfulness, image preservation, and temporal consistency metrics. Moreover, the proposed framework is lightweight and time-efficient, with a processing time of approximately one minute per video. It can generate multiple compatible edits based on a single text prompt.

**Audience:**

Yes

**Broader Impact Concerns:**

NA.

**Claims And Evidence:**

Yes

**Requested Changes:**

Please kindly refer to the weakness section.

**Strengths And Weaknesses:**

Pros:
Both quantitative and qualitative evaluations are performed. The results look promising.

Cons:

- The abstract section is not well written. The authors claim that “Recently, diffusion-based generative models have achieved remarkable success for image generation and edition. However, their use for video editing still faces important limitations.” However, the authors fail to mention what the specific limitations are, which would look confusing to the readers.

- Also, the authors claim in the abstract section that “With this framework, processing a single video only takes approximately one minute, and it can generate multiple compatible edits based on a unique text prompt.” However, the authors did not explain the equipment used for the “one minute”.

- Furthermore, I do have concerns regarding the novelty of this manuscript. It seems that the proposed framework is an ensemble of existing algorithms, such as atlas-based approaches, pre-trained text-to-image diffusion models, panoptic segmenter, and HED.

- In the experiments, only the DAVIS dataset is used to validate the effectiveness of the proposed method. The authors are suggested to consider using more datasets to make the evaluation more convincing.

---

> ### Author Response · Authors · 2024-02-02
>
> We appreciate the reviewer’s feedback and the opportunity to clarify our contributions.
>
>
> * **The abstract section is not well written. The authors claim that “Recently, diffusion-based generative models have achieved remarkable success for image generation and edition. However, their use for video editing still faces important limitations.” However, the authors fail to mention what the specific limitations are, which would look confusing to the readers.**
>
> We rewrote the abstract section to enhance its clarity and coherence. Especially, we mention the constraints of existing video editing methods with the following sentences:
>
> *However, existing diffusion based video editing models lack the ability to offer precise control over generated content that maintains temporal consistency in long-term videos. On the other hand, atlas based methods provide strong temporal consistency but are costly to edit videos on the fly and lack spatial control.*
>
> (See modified abstract in the paper)
>
> * **Also, the authors claim in the abstract section that “With this framework, processing a single video only takes approximately one minute, and it can generate multiple compatible edits based on a unique text prompt.” However, the authors did not explain the equipment used for the “one minute”.**
>
> In Section 4, titled **VidEdit Setup**, we outlined our experimental configuration, indicating the utilization of a single NVIDIA Titan RTX, a general public accessible graphic card.
>
> * **Furthermore, I do have concerns regarding the novelty of this manuscript. It seems that the proposed framework is an ensemble of existing algorithms, such as atlas-based approaches, pre-trained text-to-image diffusion models, panoptic segmenter, and HED.**
>
> We assert in this paper that the integration of an atlas method with a pre-trained image diffusion model can be highly effective for a versatile video editing. While the atlas representation guarantees temporal consistency, we show that the joint combination of relevant conditional information, i.e. text-prompt to generate the target edit, semantic mask for precise local editing and edge map for structure and motion preservation (Section 3.2 and Section 4.3) to guide the diffusion process in this space grants users precise control over the final generated content and eliminates the need for any optimization during inference - which ultimately leads to a reduced editing time.
>
> * **Only the DAVIS dataset is used to validate the effectiveness of the proposed method.**
>
> Since the DAVIS dataset serves as a widely used benchmark for evaluating video editing approaches [1], [2], [3], we opt for this dataset to compare baselines. Additionally, this dataset stands as one of the few that supplies per-frame ground-truth segmentation masks for objects in videos. We make use of these ground truth masks to evaluate our editing capacity while preserving the original video content preservation in Figure 4.
>
> [1] Wu, Jay Zhangjie, et al. “Tune-A-Video: One-Shot Tuning of Image Diffusion Models for Text-to-Video Generation.” 2023 IEEE/CVF International Conference on Computer Vision (ICCV), IEEE, 2023. Crossref, https://doi.org/10.1109/iccv51070.2023.00701.
>
> [2] Ceylan, Duygu, et al. “Pix2Video: Video Editing Using Image Diffusion.” 2023 IEEE/CVF International Conference on Computer Vision (ICCV), IEEE, 2023. Crossref, https://doi.org/10.1109/iccv51070.2023.02121.
>
> [3] Bar-Tal, Omer, et al. “Text2LIVE: Text-Driven Layered Image and Video Editing.” Computer Vision – ECCV 2022, Springer Nature Switzerland, 2022, pp. 707–23. Crossref, https://doi.org/10.1007/978-3-031-19784-0_41.

---

> ### Author Response · Authors · 2024-03-04
>
> Dear Reviewer,
>
> We reiterate our appreciation for your time. We think that your concerns can be addressed and respectfully ask you to read our response and check the changes we did in the draft. As you requested, we updated the abstract section and provided additional details to address your concerns. If possible, we respectfully ask you to engage in discussion with us if you feel your concerns have not been addressed. We are hopeful that your time allows continual discussion so you can make your final recommendation when all your concerns are addressed.

---

### Decision · Action_Editor_eMKd · 2024-03-08

**Recommendation:** Accept with minor revision

**Comment:**

The AE believes that the remaining concerns are fixable with a minor revision on the clarity of the method. The authors should make sure to take all reviewer comments into account and submit an ewer version.

**Audience:**

The paper would appeal to researchers and practitioners working on video editing.

**Claims And Evidence:**

The paper proposes an effective combination of existing algorithms for video editing.
The majority of reviewer recommend acceptance. A knowledgeable third reviewer still has concerns for the current draft, mostly citing issues around the clarity of the method.  All three agree that the claims presented are supported by the text.